# Early Effects of Extracellular Vesicles Secreted by Adipose Tissue Mesenchymal Cells in Renal Ischemia Followed by Reperfusion: Mechanisms Rely on a Decrease in Mitochondrial Anion Superoxide Production

**DOI:** 10.3390/ijms23062906

**Published:** 2022-03-08

**Authors:** Jarlene A. Lopes, Federica Collino, Clara Rodrigues-Ferreira, Luzia da Silva Sampaio, Glória Costa-Sarmento, Camila H. C. Wendt, Fernando P. Almeida, Kildare R. Miranda, Tais H. Kasai-Brunswick, Rafael S. Lindoso, Adalberto Vieyra

**Affiliations:** 1Carlos Chagas Institute of Biophysics, Federal University of Rio de Janeiro, Rio de Janeiro 21941-902, Brazil; jarlenelopes@biof.ufrj.br (J.A.L.); federica.collino@unimi.it (F.C.); clara_ferreira@biof.ufrj.br (C.R.-F.); sampaio.lu@biof.ufrj.br (L.d.S.S.); sarmento@biof.ufrj.br (G.C.-S.); camilawendt@biof.ufrj.br (C.H.C.W.); kmiranda@biof.ufrj.br (K.R.M.); 2National Center of Science and Technology for Regenerative Medicine/REGENERA, Federal University of Rio de Janeiro, Rio de Janeiro 21941-902, Brazil; tais@cenabio.ufrj.br; 3Laboratory of Translational Research in Pediatric Nephro-Urology, Department of Clinical Sciences and Community Health, University of Milan, 20122 Milan, Italy; 4National Center of Structural Biology and Bioimaging/CENABIO, Federal University of Rio de Janeiro, Rio de Janeiro 21941-902, Brazil; fepealmeida@micro.ufrj.br; 5National Center of Science and Technology for Structural Biology and Bioimaging/INBEB, Federal University of Rio de Janeiro, Rio de Janeiro 21941-902, Brazil; 6Graduate Program in Translational Biomedicine/BIOTRANS, Grande Rio University/UNIGRANRIO, Duque de Caxias 25071-202, Brazil

**Keywords:** extracellular vesicles, mesenchymal cells, proximal tubular cells, renal ischemia/reperfusion, mitochondria, anion superoxide, acellular therapy, regenerative medicine

## Abstract

Acute kidney injury (AKI) caused by ischemia followed by reperfusion (I/R) is characterized by intense anion superoxide (O_2_^•−^) production and oxidative damage. We investigated whether extracellular vesicles secreted by adipose tissue mesenchymal cells (EVs) administered during reperfusion can suppress the exacerbated mitochondrial O_2_^•−^ formation after I/R. We used Wistar rats subjected to bilateral renal arterial clamping (30 min) followed by 24 h of reperfusion. The animals received EVs (I/R + EVs group) or saline (I/R group) in the kidney subcapsular space. The third group consisted of false-operated rats (SHAM). Mitochondria were isolated from proximal tubule cells and used immediately. Amplex Red™ was used to measure mitochondrial O_2_^•^^−^ formation and MitoTracker™ Orange to evaluate inner mitochondrial membrane potential (Δψ). In vitro studies were carried out on human renal proximal tubular cells (HK-2) co-cultured or not with EVs under hypoxic conditions. Administration of EVs restored O_2_^•−^ formation to SHAM levels in all mitochondrial functional conditions. The gene expression of *catalase* and *superoxide dismutase-1* remained unmodified; transcription of *heme oxygenase-1* (*HO-1*) was upregulated. The co-cultures of HK-2 cells with EVs revealed an intense decrease in apoptosis. We conclude that the mechanisms by which EVs favor long-term recovery of renal structures and functions after I/R rely on a decrease of mitochondrial O_2_^•−^ formation with the aid of the upregulated antioxidant HO-1/Nuclear factor erythroid 2-related factor 2 system, thus opening new vistas for the treatment of AKI.

## 1. Introduction

Acute kidney injury (AKI) is one of the most severe systemic syndromes in internal medicine, with significant mortality rates, especially in intensive-care units (ICUs), where it accounts for more than 15% of the hospitalized patients [1] and 50% of those that are critically ill [2]. The economic impact is also very high; in the United States, it is estimated that it is greater than 20 billion dollars per year [3]. In the face of the controversy regarding several non-specific treatments that are offered nowadays and the severity of the outcomes [4,5], the search for therapeutic alternatives is growing worldwide. In the last few decades, replacements started focusing on cell therapies to prevent or slow the progression from AKI to chronic kidney disease (CKD). In the beginning, expectations were centered on renal resident adult stem cells [6,7], but this hope was almost totally dissipated when the existence of such cells was challenged [8,9]. Now, the use of progenitor cells is intensely focused on mesenchymal stromal cells (MSC) and the vesicles they secrete to the extracellular milieu (EVs). Recently, we demonstrated that EVs secreted by adipose MSC block the progression of cellular and molecular lesions 3 days after AKI provoked by ischemia followed by reperfusion (I/R) in rats [10].

The sudden temporary impairment of renal blood flow (ischemia) followed by restoration of circulation (reperfusion) is a central event in AKI of different etiologies [11], and intense production of anion superoxide (O_2_^•−^) occurs in the ischemic kidney followed by reperfusion (I/R) [12]. The exacerbated production of reactive O_2_ species (ROS)—together with an important inflammatory component—is a central process in the physiopathogenesis of AKI-associated lesions as a consequence of I/R, which initiates a cascade of molecular events and aberrant signaling that culminates in tubular destruction [13,14], severe impairment of renal function, and high mortality or frequent progress to CKD and end-stage renal disease [15]. Increased Bax/Bcl2 ratios and increased lipid peroxidation are key pathological molecular events resulting from the exacerbated formation of O_2_^•−^ during the I/R-induced AKI [14] because they can lead to mitochondrial damage and cellular necrosis.

The early moments of AKI onset and the response to treatments are crucial to define AKI evolution and prognosis [16]. Therefore, knowledge regarding the molecular and cellular events in these early stages can support the search for appropriate therapies and their targets, especially the mechanisms altered after the burst and during ongoing ROS formation. Considering that the effect of EVs secreted by adipose MSC, following subcapsular administration at the beginning of reperfusion, prevented or diminished tissular and functional damage 3 days after I/R [10], the present study aimed to investigate the early (24 h) EVs-induced molecular processes responsible for the late beneficial outcomes. We previously demonstrated that EVs stained with the fluorescent probe Vybrant DiD diffuse and are uniformly distributed in the kidneys 24 h after their injection [10]. We hypothesized that they could modulate processes linked to preventing oxidative damage before sustained transcriptional and translational renoprotective effects occur.

Specific objectives of this study were to investigate whether subcapsularly administered EVs, i.e., an acellular therapeutic approach, could modulate: (**i**) renal ROS production 24 h after I/R (bilateral arterial clamping for 30 min followed by 24 h of reperfusion), (**ii**) the O_2_ consumption in different respiratory states to dissect the influence of I/R and EVs on electron fluxes and ATP synthesis, and (**iii**) the transcriptional status of injury molecular markers, antioxidant enzymes, and pro-inflammatory cytokines. Measurements of plasma levels of the lesional biomarkers’ urea and creatinine aimed to determine the renal damage status. Finally, we investigated the effect of EVs on apoptosis in vitro and on electrical potential difference across the inner mitochondrial membrane potential (Δψ), using cytometry and cultures of an established lineage of human renal proximal tubule cells (HK-2) subjected to 24 h of hypoxia.

## 2. Results

### 2.1. Characterization of Extracellular Vesicles Secreted by Adipose Tissue Mesenchymal Stromal Cells

The mesenchymal stromal cells (MSC) of adipose origin (ADMSC) secreted the EVs presented in Figure 1A. They constitute a population of microvesicles (diameter above 120 nm) and exosomes (diameter ranging 50–120 nm), as demonstrated by the transmission electron microscopy (TEM) representative images. The images at higher magnification reveal a well-delimited membrane that encircles non-homogeneous content. The raw images are presented in Appendix A. In Figure 1B, the detection of surface EV markers [17] confirms the nature of the vesicles seen in panel A, with enrichment of exosome markers such as tetraspanins and endosomal-related proteins. In a previous study [10], we demonstrated the presence of the MSC markers CD73, CD90, and CD105 (but not CD146) in the EVs isolated from cultures of ADMSC (see Appendix A).

### 2.2. Mitochondria Subjected to Hypoxic Damage Are Key Targets for the Extracellular Vesicles Secreted by Mesenchymal Cells

Processes that contribute to the maintenance of the electrical potential difference across the inner mitochondrial membrane (Δψ) are key for the potential recovery of damaged cells with high metabolic demand and elevated rates of O_2_ consumption. This is the case for renal proximal tubule cells, in which mitochondrial integrity is required for the synthesis of ATP in a nephron segment where active processes of transport, mainly of Na^+^, take place at high rates [18]. Figure 2A demonstrates the normal mitochondrial morphology in HK-2 cells cultured in CTR conditions (i.e., the cells are maintained in normoxia, on the left), the appearance of pathological myelin figures in mitochondria from cells cultured in hypoxia (in the middle of the panel), and the recovery of the normal mitochondrial morphology when HK-2 cells were subsequently co-cultured with EVs (on the right). The raw images are presented in Appendix A.

The bar graph in Figure 2B quantifies the mitochondrial area, showing its decrease in hypoxia and the recovery of the CTR mean value when the renal cells were subsequently cultured with EVs.

Figure 2C shows representative images of MitoTracker™ Orange-labeled (at three different magnifications). The mitochondrial network around the nucleus is dense and intertwined within HK-2 cells in CTR (a qualitative indication of preserved Δψ; on the left) and rarified in the cells subjected to hypoxia for 24 h (in the middle). When HK-2 cells previously exposed to 1% O_2_ for 24 h were subsequently co-cultured with 1% O_2_ plus EVs for an equal period of time, the CTR mitochondrial morphology was partially recovered (on the right). Figure 2D presents the distribution of the number of events normalized to the modal value, which were recorded in flow-cytometry analysis of HK-2 cells also stained with MitoTracker™ Orange. Mean fluorescence intensity (MFI), which was also analyzed by flow cytometry, highlights the complete overlap in the intensity measured in CTR, HPX, and HPX + EVs conditions, indicating that despite the mitochondrial morphological alterations, Δψ is preserved under the hypoxia assay conditions. Figure 2E (mean ± SEM) quantifies the values of mitochondrial fluorescence intensities in arbitrary units (AU). These data suggest that organization and structure of renal mitochondria may be the target of potential beneficial effects of EVs, in early stages after hypoxic injuries.

### 2.3. Subcapsular Administration of Extracellular Vesicles Secreted by Adipose Tissue Mesenchymal Cells Restores Normal Values of Anion Superoxide Formation by Renal Mitochondria after Ischemia/Reperfusion

The rates of O_2_^•−^ formation by isolated renal mitochondria in different respiratory states were measured in SHAM, I/R, and I/R + EVs rats after 24 h of reperfusion. Figure 3 presents representative recordings of resorufin fluorescence variations after successive additions of substrates, uncoupler, and inhibitors to mitochondrial suspensions. These recordings correspond to H_2_O_2_ formation after dismutation of the O_2_^•−^ that is produced as a consequence of premature transfer to O_2_ of the electrons removed during oxidation of endogenous substrates and succinate. Comparison of Figure 3A,B shows that the velocity of O_2_^•−^ formation increases by 50% after addition of succinate (traces between peaks 1 and 2) in mitochondria isolated from I/R rats, which returned to the SHAM value when EVs were injected at the beginning of reperfusion (Figure 3C). Quantification of the velocities when mitochondria oxidize succinate in non-phosphorylating conditions is given in Figure 3D. It can be seen that EVs treatment totally recovers the control formation of H_2_O_2_.

After adding 0.1 mM ADP, a condition in which phosphorylation to ATP occurs at low velocity, there was no difference among the three groups (average of 1000 pmol H_2_O_2_ × mg^−1^ × min^−1^ over the entire period) (data not shown). When electron transfer and ATP formation were accelerated by the addition of a higher ADP concentration (1 mM) (Figure 4A), H_2_O_2_ formation decreased in the three groups (compare with Figure 3D) and the 50% stimulation in the I/R group with respect to SHAM appeared again, as well as the recovery of the SHAM velocity in the I/R + EVs group. A similar profile of stimulation/recovery, but at higher levels, was encountered when ATP synthesis—and respiration—was blocked by oligomycin (Figure 4B). When the respiration became accelerated and the H^+^ electrochemical gradient dissipated by the addition of the uncoupler FCCP (Figure 4C), the H_2_O_2_ formation decreased in all groups. Still, the profile of increase by I/R and total normalization by EVs persisted. Finally, when electron fluxes were greatly diminished by inhibition of Complex III by Antimycin A, the rate of H_2_O_2_ formation increased by 100% with respect to SHAM, and the decrease promoted by EVs was partial (Figure 4D).

An interesting feature was encountered when H^+^ leak, one of the most important mechanisms of antioxidant defense in mitochondria [19], was investigated. Proton leak in mitochondria (i.e., the return of H^+^ from the interspace to the matrix through pathways different from the F_o_F_1_-ATPsynthase) can be estimated from the difference between the QO_2_ in the presence of oligomycin and the residual QO_2_ after inhibition of Complex III by Antimycin A [20,21,22]. Figure 5 shows that H^+^ leak was similar in the three groups and, therefore, that the ischemic episode followed by reoxygenation did not affect the pathways through which H^+^ flows back to the matrix through pathways different from the F_o_F_1_-ATPsynthase.

### 2.4. Twenty-Four Hours after Ischemia Followed by Reperfusion, the Acute Renal Lesions Still Persist

To see whether the early normalization of the redox status of renal cortical mitochondria was associated with amelioration of the acute lesions provoked by the ischemia followed by reperfusion, we investigated the expression of two biomarkers of kidney damage 24 h after the injury. They were *Kidney Injury Molecule-1* (*KIM-1*) and *Neutrophil Gelatinase-Associated Lipocalin* (*NGAL*), which are considered sensitive and specific markers of proximal tubule lesions [23,24]. *KIM-1* was barely detectable in SHAM conditions and increased approximately 20-fold in the renal cortex corticis of I/R rats, a level that remained unmodified in the group I/R + EVs (Figure 6A). As in the case of *KIM-1*, the expression of *NGAL* was very low in SHAM rats, increasing more than 15-fold after I/R without influence of EVs treatment (Figure 6B). The persistent pathological dysfunction of AKI is confirmed by the elevated levels of urea (87.8 ± 2.0 mg/dL vs. 57.8 ± 1.0 mg/dL in SHAM; *p* = 0.0017) and creatinine (0.47 ± 0.01 mg/dL vs. 0.28 mg/dL in SHAM; *p* = 0.0001) in blood, which were not modified by EVs administration in the 24 h window of lesion (100.3 ± 10.4 mg/dL for urea, NS with respect to I/R, *p* = 0.0001 vs. SHAM; 0.52 ± 0.05 mg/dL for creatinine, NS with respect to I/R, *p* < 0.0001 vs. SHAM) (one-way ANOVA, followed by Tukey’s test; *n* = 5–7).

Since hypoxia and exacerbated ROS formation can trigger and positively feed inflammatory processes in kidney tubule cells and their interstitial surroundings [13], the next step was to study the gene expression of two pro-inflammatory cytokines: *Interleukin-6* (*IL-6*) (Figure 7A) and *Tumor Necrosis Factor-alpha* (*TNF-α*) (Figure 7B). Both cytokines were remarkably upregulated after ischemia followed by 24 h of reperfusion to 700% and 100% higher levels, respectively, with respect to SHAM values. The effects of the EVs administration were different for each cytokine: *IL-6* dropped by half compared to I/R and *TNF-α* remained unmodified in I/R + EVs rats, probably due to different influences of redox environment alterations in the release of these cytokines. This point will be discussed below.

### 2.5. The Early Anti-Oxidative Responses after EVs Administration Co-Exist with Unmodified Transcriptional Activation of Key Enzymes Involved in Redox Regulation

It is well known that enzyme-catalyzed antioxidant mechanisms control the balance between cell death and survival in acute renal lesions [25]. Thus, we investigated the gene expression: (**i**) of a master regulator of cellular redox homeostasis and mitochondrial function [26,27], the *heme oxygenase-1* (*HO-1*), and (**ii**) of two enzymes that catalyze the sequential split of H_2_O_2_ to H_2_O and O_2_ after its formation by dismutation of O_2_^•−^: *catalase* (*CAT*) and *superoxide dismutase* (*SOD*), respectively. The expression of *HO-1* in proximal tubules increased by 300% after I/R compared to SHAM. The upregulated levels remained unmodified by EVs injection (Figure 7C), evidence that I/R triggered an antioxidant mechanism of defense that could participate in the early restoration of the mitochondrial redox status presented in Figure 3 and Figure 4, without any influence of the EVs components. The profile was the opposite for *CAT* (Figure 7D) and *SOD* (Figure 7E) transcription, which was downregulated in I/R again without any EVs effects, indicating that I/R partially inhibited transcriptional processes unable to be restored—in the early phase of the damage—by the factors that the vesicles carry.

### 2.6. Apoptotic Processes Are Partially Reversed When Renal Proximal Tubule Cells Are Co-Cultured with Extracellular Vesicles after Hypoxia

We investigated whether EVs can prevent cell death in a lineage of kidney proximal tubule cells besides their beneficial influence on the increased redox stress. HK-2 cells subjected to hypoxia for 24 h were analyzed by Annexin/PI staining (Figure 8). Hypoxia, which mimics ischemia in vitro, increased the number of apoptotic cells, especially of those in late-stage, i.e., of ANX^+^PI^+^ cells (by approximately threefold; Figure 8A,B,E), and increased by 100% the number of cells in an early stage of apoptosis (ANX^+^PI^−^) (Figure 8A,B,D). The co-culture with EVs in the reoxygenation phase significantly decreased the process of early cell death (Figure 8C,D), but not that of late apoptosis (Figure 8C,E). When the number of total apoptotic cells (early and late stage) (Figure 8F) was considered, the mean value of the group (HPX + EVs) (6.2%) was statistically similar to those of the groups CTR (4.0%) and HYX (8.3%).

## 3. Discussion

The main finding in this study was that subcapsular administration of MSC-derived EVs at the moment of reperfusion after ischemia totally recovers the normal ROS formation by mitochondria from renal proximal tubule cells in all functional respiratory states. Recovery of basal and controlled production of ROS is required for proper mitochondrial function, including ATP synthesis and energy delivery for a varied ensemble of cell processes, especially ion transport [28], and this was achieved within 24 h after I/R by the EVs that diffused from the renal subcapsular region to the entire parenchyma, as we recently demonstrated by in vivo biofluorescence approaches [10].

We found that electron leak, i.e., the premature transfer of electrons to O_2_ forming O_2_^•−^ [29], is the major early mitochondrial molecular event after non-septic AKI caused by I/R. The rate of O_2_^•−^ production rose by more than 50% after mitochondrial energization by addition of succinate, the respiratory substrate of Complex II (Figure 3 and Figure 4), even though this complex per se is not a source of O_2_^•-^ formation [30,31]. This substrate was chosen for the following reasons. First, during ischemia, the destruction of tubule cells releases fatty acids that are oxidized, forming FADH_2_ from FAD at the level of Complex II; when reoxygenation occurs, the electrons it carries are transferred to Complexes III and IV. At the level of Complex III, there is an intense production of O_2_^•−^ [30] that can be carried to the cytosol by voltage-dependent anion channels [32]. The second reason for choosing succinate was that O_2_^•−^ formation at the level of Complex I resulted from reverse transfer of electrons from FADH_2_ [33], thus activating a new site for the mitochondrial generation of ROS. The importance of the EVs-induced total blockade of the uncontrolled O_2_^•−^ production and tissue damage emerges from the observation that damage of cytosolic structures is propagated to intact mitochondria, which amplifies the vicious circle of mitochondrial destruction [34].

Of particular interest is the effect of I/R—and of EVs—when O_2_^•−^ formation is assayed in phosphorylating conditions after the addition of 1 mM ADP (Figure 4A). The production of O_2_^•−^ after I/R decreased to less than 15% when compared with the values obtained in the presence of succinate alone (Figure 3D), as a consequence of the dissipation of the Δ_eleq_H^+^ during the simultaneous ATP synthesis [33,35]. The complete recovery of control rates of O_2_^•−^ production points to the beneficial influence of EVs in maintaining ATP synthesis and recovering the lower velocity of O_2_^•−^ formation, thus avoiding the amplified damage of I/R through the impairment of ATP synthesis. The similarity between the profiles in phosphorylating conditions and with FCCP (uncoupled) (Figure 4A,C) supports the view that dissipation of Δ_eleq_H^+^ is the mechanism on which relies the decrease in the overall rate of O_2_^•−^ formation. The 50% increase of O_2_^•−^ formation after I/R could be attributed to the intensification of the pro-oxidant microenvironment and electron leak in the vicinity of the Fe.S-containing 14 central subunits of Complex I [36], which alternate between successive cycles of oxidation/reduction during the catalysis of electron transport [37]. Possibly, the premature transfer of electrons to O_2_ occurs to a lesser extent in Complex III, conceivably because it contains fewer Fe.S centers in its dimeric structure [38].

The view of selective oxidative damage of the electron transport system as a critical early mechanism for mitochondrial dysfunction is reinforced by the observation that O_2_^•−^ production increases by more than 80% in the I/R group in comparison with SHAM when phosphorylation is blocked by oligomycin (Figure 4B), whereas H^+^ leak—which is estimated by the QO_2_ in this condition and is considered a central antioxidant defense [19,22]—remains unmodified (Figure 5). Since increased O_2_^•−^ enhances H^+^ leak, thus favoring uncoupling, and uncoupling decreases O_2_^•−^ [39], it may be that the lack of effects of I/R observed in Figure 5 results from protective feedback mutually involving the two processes.

The modest partial recovery of the rate of O_2_^•−^ formation in the group I/R + EVs when compared to SHAM, as well as the high acceleration caused by I/R when the ETS is blocked by Antimycin A at the level of Complex III (Figure 4D), are probably due to the complexity of the pathways for the electron fluxes when this inhibition occurs. When Complex III is blocked, the electrons flow from succinate toward Complex IV through the alternative pathway that requires high spatial organization. It involves reverse transfer of electrons to Complex I [33] that is part of the supercomplex I_1_ + III_2_ + IV_1_ [40,41,42], followed by electron tunneling through the mobile pool of coenzyme Q towards Complex IV. It is likely that under intense pro-oxidant activity and lipid peroxidation, this pool is easily destructured, favoring electron leak, increasing further O_2_^•−^ formation and disrupting repair mechanisms.

The recovery of the normal mitochondrial redox status by EVs seems to be a very early process that is not accompanied by an overall improvement of the injuries caused by I/R. Neither gene expression of *KIM-1* nor that of *NGAL*, two key lesional biomarkers of acute renal injury, was modified by EVs (Figure 6), thus confirming that intense tubular lesions persist. Since 72 h after I/R, the expression of the tubular lesional biomarker *KIM-1* was strongly downregulated in rats given EVs [10], and this was accompanied by almost normalization of the high Bax/Bcl2 ratio, an indicator of mitochondrial recovery [43], it may be proposed that early preservation of a physiological redox mitochondrial microenvironment during the first 24 h of reperfusion after ischemia helped to avoid elevated lipid peroxidation and, therefore, activation of Bax-related proteins with later apoptosis. In addition, this ensemble of events occurred in the middle of extensive tubular damage, as demonstrated by the maintenance of I/R high levels of *KIM-1* and *NGAL* after administration of EVs. In the case of *NGAL*, an opposite view deserves consideration. Its upregulated expression could be viewed as indicative of an antiapoptotic response that can help later recovery, as proposed for an in vivo endotoxin-induced model of AKI in rats [44].

Tubular damage is accompanied by a persistent inflammatory response, as evidenced by the elevated levels of the cytokines *IL-6* and *TNF-α* (Figure 7), whose expression increased in several models of AKI [45,46]. The elevated level of pro-inflammatory *TNF-α*, which remained unmodified in rats receiving EVs, indicates immune infiltration [47] and is probably responsible for the only partial decrease of *IL-6* levels [48]. The amount remaining after 24 h could contribute to reducing lipid peroxidation and oxidative damage [49]. Possibly, the partial selective early decrease of the master regulator *IL-6* [50] by EVs results from the release of anti-inflammatory factors contained within the EVs that were demonstrated in our previous proteomic studies [10].

The decrease in O_2_^•−^ promoted by EVs in all respiratory states in the first 24 h was not due to transcriptional upregulation of the enzymes that, sequentially, catalyze the formation of H_2_O_2_ from O_2_^•−^ (*SOD*) and its conversion to H_2_O and O_2_ (*CAT*), because their levels remain the same as those encountered in I/R (Figure 7D,E). These enzymes became downregulated as the ROS production increased in I/R [51,52,53], thus worsening the prognosis of tubular damage. It may be that the sudden decrease in the pO_2_ during the ischemia together with the formation of O_2_^•−^ during reperfusion resulted in the shutdown of transcription promoters [54], as demonstrated for the CAT from normal and tumoral cells from different origins, together with the destabilization of preexisting mRNA [25].

Of particular relevance is the upregulation of *HO-1* by I/R, which was not modified in I/R that received EVs (Figure 7C). The mRNA levels 24 h after I/R are similar to those encountered after 72 h [10], when immunohistochemistry analysis revealed small areas positively stained for HO-1. This correlation may indicate that the upregulation of this antioxidant gene is an essential part of the early intrinsic response of kidney cells facing the oxidative stress caused by I/R. This protective response is associated to a previous step of activation of the Nuclear factor (erythroid-derived 2)-like 2 (Nrf2) and translocation to the nucleus, which is ensured by rapid, numerous, and interacting pathways [55]. Due to the role of HO-1 in the regulation of processes such as inflammation and apoptosis [27], its early upregulation without further modification by EVs appears to be located at a crossroad between apoptosis and survival, collaborating with the rapid EVs-mediated protective antioxidant mechanisms in mitochondria discussed above. This view is supported by recent observations demonstrating that administration of anti-ischemic drugs such as meldonium increases the expression of Nrf2 with simultaneous decrease of lipid peroxidation and of the Bax/Bcl2 ratios in a model of I/R in rats [14].

The images and the bar graph in Figure 8 revealed that EVs transferred to renal cells—at least in part—the molecular machinery able to stimulate mechanisms of repair previously characterized by proteomic studies [10]. It is clear, however, that the overall benefit of EVs is partial, because the sum of cells in early and late apoptosis that were cultured with EVs was not different from either the CTR group or the HPX group. The antioxidant effects evidenced by the decrease in O_2_^•−^ formation and upregulation of the *HO-1* transcription, early protective mechanisms of EVs, resulted in in a partial decrease in apoptosis in renal cells cultured with EVs after they were subjected to extreme anoxia. It is worth mentioning that the upregulation of these mechanisms is related to cell survival and proliferation [56,57]. One of the key factors shuttled by EVs secreted by MSC is catalase, whose inactivation suppresses the beneficial effects of EVs [58]. Its release after diffusion of EVs into renal parenchyma [10] may be responsible for suppressing the excess of O_2_^•−^ formed during I/R.

## 4. Materials and Methods

### 4.1. Ethical Statement

All experimental procedures were approved by the Committee of Ethics in Animal Experimentation of the Federal University of Rio de Janeiro (CEUA) (protocol N° A02/16-61-15) and performed under the Committee’s guidelines, following the Uniform Requirements for Manuscripts Submitted to Biomedical Journals. The animal study is reported in accordance with ARRIVE guidelines [59].

### 4.2. Isolation and Characterization of Extracellular Vesicles Secreted by Adipose Mesenchymal Stromal Cells

Human adipocyte mesenchymal stromal cells (MSC) were purchased from Lonza (Basel, Switzerland). They were cultured in dishes with Dulbecco´s modified Eagle medium (DMEM) (Life Technologies, Carlsbad, CA, USA) at 37 °C under an atmosphere of 5% CO_2_ in air. When cells attained 80% confluence, they were washed twice with phosphate-buffered saline (PBS) at pH 7.2 and starved for 14–16 h in Roswell Park Memorial Institute (RPMI) medium without phenol red (Thermo Fisher Scientific, Waltham, MA, USA). The culture medium was then centrifuged at 1713× *g* for 20 min at room temperature to remove cell debris and apoptotic bodies, and the supernatant was centrifuged at 100,000× *g* for 2 h at 4 °C. The recovered sediment contained exosomes and microvesicles (40–100 nm and 100–500 nm, respectively; Figure 1A), i.e., a heterogeneous population of EVs (resuspended in 200 µL RPMI containing 1% dimethyl sulfoxide and immediately stored at −80 °C).

For the morphological characterization of the vesicles by transmission electron microscopy, the pellets obtained as above were resuspended in PBS (pH 7.2) and adhered onto glow-discharged formvar-coated copper grids (300 mesh; EMS, Hatfield, PA, USA) for 10 min. Excess solution was removed using filter paper (Whatman No 1; Merck, Darmstadt, Germany). Then, the grids were negatively stained with 1% (*w/v*) aurothioglucose in water for 30 s and dried with the same filter paper. The samples were observed on a Tecnai-Spirit electron microscope (Thermo Fisher Scientific, Waltham, MA, USA) operating at 120 kV and equipped with a 2K camera (Veleta, Olympus, Münster, Germany).

The EVs surface antigens CD9, CD63, TSG101, and CD81, were assayed by Western blotting (Figure 1B) using the corresponding antibodies. CD9 (#ab92726, anti-rabbit, at 1:100 dilution) was from Abcam (Cambridge, United Kingdom). CD63 (#sc-5275), TSG101 (#sc-136111), and CD81 (#sc-70803) were from Santa Cruz Biotechnology (Dallas, TX, USA) (anti-mouse in the 3 cases at 1:100 dilution). The secondary antibodies were anti-rabbit (1:2500; GE Healthcare, Cheshire, United Kingdom) and anti-mouse (1:5000; GE Healthcare). The procedures for immunodetection were as recently described [60], except that the protein immunosignals were detected using the ImageQuant LAS 4000 system (GE Healthcare, Chicago, IL, USA).

### 4.3. Animals and Treatments

We utilized adult male Wistar rats weighing 200–300 g, available at the Central Animal Facilities Health Sciences Center, Federal University of Rio de Janeiro (Rio de Janeiro, Brazil). The animals were housed in a room maintained at 23 ± 1 °C on a 12:12 h light–dark cycle and allowed to acclimatize for 1 week, with free access to a commercial chow for rats (Purina Agribrands, Paulínia, Brazil) and filtered tap water. As stated above, the experimental protocols followed the guidelines and were approved by the local ethical committee.

The rats were randomly divided into 3 groups: (**i**) sham operated (SHAM), (**ii**) ischemia/reperfusion (I/R), and (**iii**) I/R that received subcapsular injections of 2 × 10^9^ EVs (in 150 µL PBS) at the moment of reperfusion (I/R + EVs). The group I/R received the same volume of saline. The I/R protocol was as previously described with slight modifications [61]. Briefly, the rats were anesthetized with 0.2 mL ketamine 10% plus 0.1 mL xylazine 2% (Syntec, Santana de Parnaíba, Brazil) intraperitoneally, and the anesthesia was confirmed by pressing the hind legs. The abdominal cavity was opened, the renal pedicles were dissected, and the renal arteries were clamped for 30 min using a stainless-steel clamp. The pedicle was smoothly manipulated in the SHAM group after visualization without dissection.

The rats received a xylocaine topical ointment and were placed in individual cages under the same previous conditions for 24 h. They were sacrificed by decapitation, and both kidneys were immediately removed. For the experiments intended for studies of ROS formation and O_2_ consumption, the mitochondria of the external portion of the cortex (cortex corticis) were isolated as previously described [61] with slight modifications. In this segment of the renal tissue, more than 90% of the cell population corresponds to proximal tubules [62]. Briefly, the cortical fragments were washed twice using a solution containing 250 mM sucrose, 10 mM HEPES-Tris (pH 7.4), 2 mM Na_2_EDTA, and 0.15 mg/mL trypsin inhibitor (Sigma-Aldrich, St. Louis, MO, USA), which was used in the following steps. After gentle manual homogenization using a glass Potter-Elvehjem homogenizer (Merck), the total homogenates were centrifuged at 4 °C. First, at 600× *g* for 5 min to sediment intact cells, cell debris, and nuclei, recovering the supernatants that were immediately centrifuged for 10 min at 12,000× *g* to obtain mitochondria-enriched sediments. These sediments were resuspended in 5 mL of the above-described solution and centrifuged again at 12,000× *g* to obtain a washed pellet containing mitochondria, finally resuspended in 300 µL of the above solution and immediately used. For the experiments intended for mRNA analysis and relative quantitative expression, the whole cortex region was processed as recently described [10].

### 4.4. Renal Cells Cultures and Hypoxia/Reoxygenation Protocol

Immortalized cells from human kidney proximal tubules (HK-2 lineage; ATCC, Manassas, VA, USA) were cultured in 6-well plates (5 × 10^5^ cells per well) in Keratinocyte Serum-Free Medium (K-SFM) (Thermo Fisher Scientific, Waltham, MA, USA) supplied with 5% (*v/v*) bovine fetal serum (Gibco™, Thermo Fisher Scientific, Waltham, MA, USA), under an atmosphere of 5% CO_2_ in air at 37 °C, until they reached 80% confluence. The cells were then divided into 2 lots and cultured in serum-free DMEM, under normoxia (19% O_2_, 5% CO_2_ in air) or hypoxia (1% O_2_, 5% CO_2_ in N_2_), at 37 °C for 24 h. Next, the plates were cultured for 24 h at 37 °C under normoxia. Some of the cells previously subjected to 24 h hypoxia conditions were co-cultured with 2 × 10^9^ EVs. Thus, we obtained 3 groups: (**i**) CTR, 24 h normoxia → 24 h normoxia; (**ii**) HPX, 24 h hypoxia → 24 h normoxia; (**iii**) HPX + EVs, 24 h hypoxia → 24 h normoxia in co-culture with EVs.

### 4.5. Flow Cytometry

The 3 populations of HK-2 cells were detached by adding 1 mL of PBS containing 0.05% (*w/v*) porcine pancreatic trypsin (Trypsin-EDTA, Thermo Fisher Scientific, Waltham, MA, USA). The cell suspensions were centrifuged at 321× *g* for 5 min at room temperature, and the pelleted cells were resuspended in 100 µL of a solution containing 10 mM HEPES-Tris (pH 7.4), 140 mM NaCl, and 2.5 mM CaCl_2_ (solution C from Annexin V kit, Thermo Fisher Scientific, Waltham, MA, USA). The samples (50 µL of the suspension) were supplied with 3 µL of Annexin V (solution A from Annexin V kit) and 3 µL of a solution containing 0.15 mM propidium iodide (10-fold diluted solution B from Annexin V kit). The cytometry analyses were carried out immediately after 15 min at room temperature using an Accuri™ C6 flow cytometer (Accuri Cytometers Inc./Becton Dickinson, Franklin Lakes, NJ, USA). The remaining 50 µL of the cell suspensions were supplied with 50 nM MitoTracker™ Orange (Molecular Probes, Eugene, OR, USA) diluted in DMEM, incubated for 15 min, mixed with 700 µL RPMI, centrifuged at 321× *g* for 5 min, and washed 3 times with this solution. The final pellet was resuspended in 300 µL RPMI, and immediately observed in the same Accuri™ C6 flow cytometer.

### 4.6. Analysis of Mitochondrial Morphology and Qualitative Evaluation of the Electrical Potential Difference across the Inner Mitochondrial Membrane from Renal Cells in Co-Culture with Extracellular Vesicles

HK-2 cells were fixed in 2.5% (*v/v*) glutaraldehyde with 4% (*v/v*) formaldehyde in 0.1 M cacodylate buffer (pH 7.2) for 24 h. Following a wash in cacodylate buffer 0.1 M, the samples were post-fixed in 1% OsO_4_ (*w/v*) plus 0.8% (*w/v*) K_4_(CN)_6_Fe in 0.1 M cacodylate buffer for 40 min, dehydrated in acetone series, and embedded in epoxide resin (Embed 812 Resin, EMS, Hatfield, PA, USA). Ultrathin sections (70 nm) were obtained using a Leica EM UC7 ultramicrotome (Leica, Wetzlar, Germany), collected onto 200 mesh copper grids (EMS), and stained for 20 min in 5% (*w/v*) uranyl acetate and 5 min in Reynolds’ lead citrate. After, the samples were observed on a Tecnai-Spirit electron microscope (FEI Company, Eindhoven, Netherlands) operating at 120 kV, equipped with a 2k CCD camera (Veleta, Olympus). Profiles of mitochondria in randomly selected cells were evaluated with ImageJ software (U.S. National Institutes of Health, Bethesda, MD, USA), and we compared the average area among the different experimental groups. Due to the polymorphic shape of the cells, we did not measure the volumetric density.

HK-2 cells were also used to qualitatively investigate the mitochondrial energetic state—represented by the transmembrane potential Δψ—after hypoxia and the influence of co-culture with EVs, as recently described [59]. HK-2 cells from the three experimental groups described above were placed in plates of 24 wells, incubated with the fluorophore MitoTracker™ Orange (Molecular Probes), and assayed as follows for immunofluorescence visualization. Since the HK-2 cells do not adhere to glass, the coverslips at the bottom of the plates were washed with 400 µL Attachment Factor Cascade Biologics™ AF (Thermo Fisher Scientific, Waltham, MA, USA) and immediately dried at 37 °C for 30 min. The cells (8 × 10^4^ per well, suspended in 500 µL DMEM without serum) were placed onto 13 mm diameter coverslips deposited at the bottom of the wells, washed twice with PBS, and supplied with 500 µL of the fluorophore diluted in a modified Krebs solution containing 120 mM NaCl, 4 mM KCl, 1.4 mM MgCl_2_, 2.5 mM CaCl_2_, 6 mM glucose, and 10 mM HEPES (pH adjusted to 7.4 with Tris), and incubated for 20 min at 37 °C. Then, the cells were washed 3 times for 5 min with the same solution and fixed with paraformaldehyde 4% (*w/v*) for 15 min at room temperature, sheltered from the light. After removal of the fixative, the cells were rewashed 3 times for 5 min using the same solution. The coverslips were carefully removed from the plate with the aid of a small forceps, supplied with 20 µL DAPI (5 µg/mL in a PBS-glycerol 40% (*v/v*) solution at pH 7.4 adjusted with Tris). The coverslips were mounted onto glass slides, sealed with colorless nail polish, and stored at −20 °C for 24 h. The images were collected using AxioVision 4.8.2 software in an ApoTome microscope (ApoTome Axion Imager.M2, Carl Zeiss Inc., Jena, Germany) at 554 nm (excitation) and 576 nm (emission).

### 4.7. Time Course of Reactive O_2_ Species Formation

The generation of mitochondrial O_2_^•−^ was quantified by measuring the changes in fluorescence when the fluorescent resorufin is formed by the peroxidase (2 U/mL)-catalyzed oxidation of the Amplex Red™ probe (5 mM, Thermo Fisher Scientific, Waltham, MA, USA), after dismutation of O_2_^•−^ into H_2_O_2_ in the presence of excess superoxide dismutase (60 U/mL, Sigma-Aldrich), as recently described [63]. Generation of O_2_^•−^ was evaluated at different respiratory states after substrate–uncoupler–inhibitor–titration (SUIT): (**i**) 10 mM succinate (substrate for Complex II); (**ii**) 0.1 mM ADP; (**iii**) 1 mM ADP; (**iv**) 0.2 µg/mL oligomycin (to inhibit ATP formation and to stimulate H^+^ leak); (**v**) 1.5 µM carbonyl cyanide-4-(trifluoromethoxy) phenylhydrazone (FCCP) to completely dissipate the H^+^ gradient; (**vi**) 2.5 µM antimycin to inhibit Complex III. The basic reaction medium before SUIT contained: 10 mM HEPES-Tris at pH 7.4, 320 mM mannitol, 4 mM KH_2_PO_4_, 4 mM MgCl_2_, 0.08 mM Na_2_EDTA, 1 mM EGTA (free acid), and 1 mg/mL fatty acids free BSA. The final pH was adjusted to 7.4 by adding Tris; the mitochondrial protein concentration was 0.1 mg/mL. Recordings were acquired at 563 nm (excitation) and 587 nm (emission). Calibration curves were obtained using successive pulses of 10 nM H_2_O_2_, thus allowing the conversion of arbitrary fluorescence units into pmol H_2_O_2_/mg per min. The stoichiometry 1 O_2_^•−^: 1 H_2_O_2_ allowed quantification of O_2_^•−^ formation rate.

### 4.8. Oxygen Consumption by Isolated Mitochondria from Kidney Proximal Tubule Cells

We measured oxygen consumption (QO_2_) by isolated mitochondria using a high-resolution O_2_ electrode (Oxygraph-2K, OROBOROS Instruments, Innsbruck, Austria) at 37 °C in the solution described above for mitochondrial O_2_^•−^ determination, except that the components required to detect H_2_O_2_ formation were omitted, and 0.06 mg/mL mitochondrial protein was used. The QO_2_ assays were run in parallel with H_2_O_2_ assays. All mitochondrial preparations were assayed to determine the respiratory control ratio (RCR) and, therefore, to evaluate the coupling between electron fluxes and ATP synthesis. The RCR was calculated from the ratio between the QO_2_ in the presence of 1 mM ADP and the QO_2_ after the addition of oligomycin. Despite differences in the absolute values of QO_2_ (lower in I/R mitochondria), the RCR averaged 3.0 in all groups, with an interassay variation coefficient of ~10%. The QO_2_ after successive additions of oligomycin and Antimycin A was used to estimate the H^+^ leak from the mitochondrial space back to the matrix [20,21,22].

### 4.9. RNA Isolation, Reverse Transcription and Real Time Quantitative Polymerase Chain Reaction (qRT-PCR)

Small fragments of kidney cortex were suspended in 500 µL of Lysis/binding buffer (miRNA isolation kit *mir*Vana™, Thermo Fisher Scientific, Waltham, MA, USA) in RNAse-free microtubes and homogenized using the dissociator TissueLyser LT provided with a 5 mm diameter bead (Qiagen, Hilden, Germany). After 7 min of intense agitation, 400 µL of the homogenate was mixed with 40 µL of miRNA homogenate additive (miRNA isolation kit *mir*Vana™), vigorously vortexed, and then placed on ice for 10 min. The suspensions were then supplied with the same volume of chloroform, vortexed again at 12,000× *g* for 5 min at room temperature, to recover supernatants of 1 mL that were mixed with 100% ethanol (1.25 mL ethanol:1 mL sample), gently homogenized with a micropipette and centrifuged at 12,000× *g* for 15 s at room temperature using microtubes supplied with the filter provided by the kit (miRNA isolation kit *mir*Vana™). After removing the liquid, the filters were first washed with 700 µL of the miRNA wash solution 1, centrifuged for 15 s at 12,000× *g* at room temperature, and then washed twice using 500 µL of the miRNA wash solution 2/3 (both solutions from the kit mentioned above). Finally, the filters were immersed in 100 µL of RNAse-free H_2_O at 95 °C, immediately centrifuged for 30 s at 13,000× *g,* and the liquids, after removal of the filters, were stored at −80 °C.

The mRNAs were obtained using 10 µL of the High-Capacity cDNA Reverse Transcription kit (Applied Biosystems™, Foster City, CA, USA)—with components freshly mixed—and 10 µL samples containing 20 ng/µL mRNA. The RNA of these samples was quantified using the NanoDrop™ ND-1000 (Thermo Fisher Scientific, Waltham, MA, USA), as recently described [10]. The cDNAs were synthesized in the thermocycler PCR Thermal Cycler (Applied Biosystems™), with cycles of 10 min at 25 °C, 120 min at 37 °C, 5 min at 85 °C, and 1 min at 4 °C, before storage at −20 °C. Negative controls without reverse transcriptase were carried out in parallel with each run.

The reverse transcription followed by the real-time quantitative polymerase chain reaction was performed in a single step using 10 µL of Power SYBR Green® PCR Master Mix (Applied Biosystems™) and 10 µL of the cDNA-containing solution and the primers (0.25 ng/µL and 100 nM, respectively). The sequence-specific oligonucleotides [10] were from Eurofins Genomics (Ebersberg, Germany) (Table 1). Their amplifications were followed using the ViiA™ 7 Real-Time System (Applied Biosystems™) after a stage of 10 min to reach 95 °C, followed by cycles of 15 min at 95 °C, 60 min at 60 °C, and 15 min at 95 °C.

### 4.10. Statistical Analysis

The mean values of the different parameters investigated in the 3 experimental groups of rats and in the 3 HK-2 cells assays were analyzed using one-way ANOVA followed by Tukey’s test. In the case of RQ data, the mean of each SHAM group was taken as 1.0, and the individual values from the 3 groups were expressed as a fraction or as a multiple of this value [63,64,65]. This allowed the SEM of the unitary value of SHAM reference to be calculated.

## 5. Conclusions

We demonstrated that the early and central mechanism by which EVs protect renal structure and function after I/R is the decrease of O_2_^•−^ production to control levels and, therefore, the normalization of the mitochondrial redox environment. The antioxidant components of EVs are central in the preservation mechanisms that, with the aid of the upregulated antioxidant HO-1, depress early processes of mitochondrial damage and cell death after I/R. The proposed molecular events elicited by EVs are depicted in Figure 9.

## Figures and Tables

**Figure 1 ijms-23-02906-f001:**
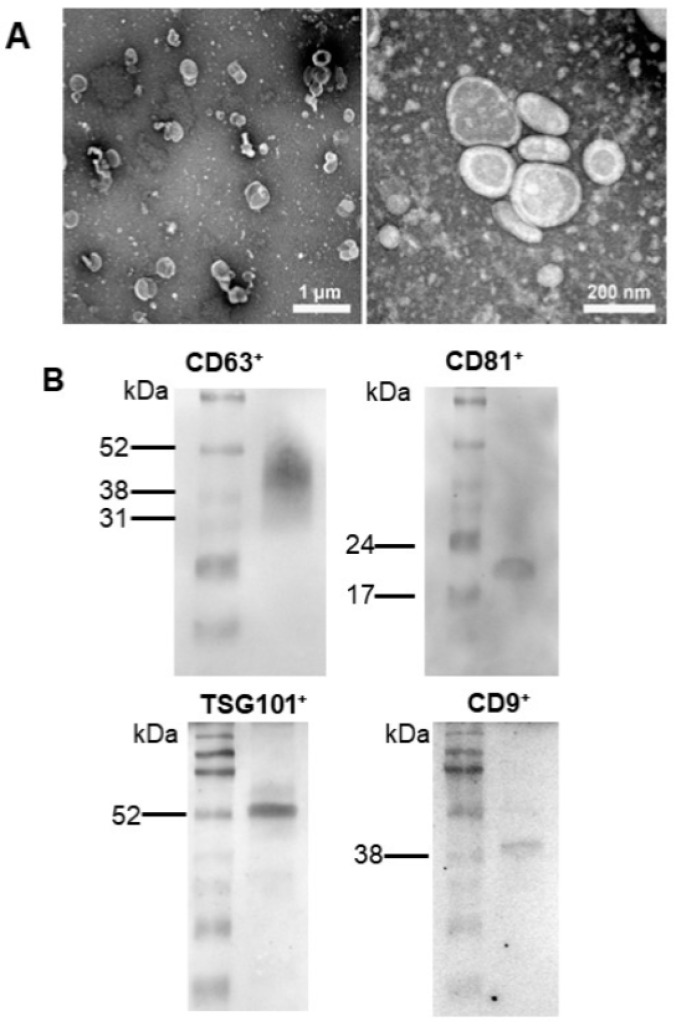
Characterization of extracellular vesicles (EVs) secreted by adipose mesenchymal cells. (**A**) Morphological characterization. Transmission electron microscopy (TEM) at the two indicated magnifications. (**B**) Characterization by using vesicular surface antigens. The immunodetections show that the EVs, whose images are presented in (**A**), are CD9^+^, CD63^+^, TSG101^+^, and CD81^+^, as indicated.

**Figure 2 ijms-23-02906-f002:**
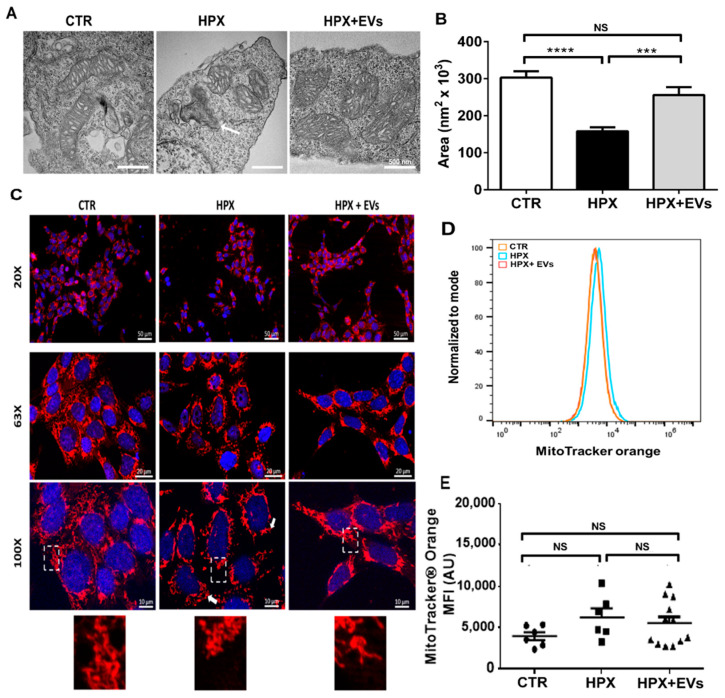
Evaluation of structural mitochondria damage after AKI and EVs treatment. (**A**) Transmission electron microscopy images of mitochondria from HK-2 cells in CTR, HPX, and HPX + EVs conditions, as indicated above the panels. Arrowhead in HPX points to a typical myelin figure. (**B**) Quantification of the mitochondrial areas. Bars represent mean ± SEM (*n* = 34, 37, 30 areas for CTR, HPX, and HPX + EVs groups, respectively). (**C**) Representative fluorescence images of MitoTracker™ Orange (red areas) to evaluate mitochondrial Δψ in CTR, HPX, and HPX + EVs. Nuclei are stained in blue (DAPI). Magnifications are indicated on the left side of the panels. The insets in the 100× magnification pictures are presented just below the panels. (**D**) The percentage of events in cytometry analyses (normalized to modal value) at the arbitrary fluorescence intensity units indicated on the abscissa. The small colored squares indicate the experimental conditions: orange, CTR; blue, HPX; red, HPX + EVs. The histograms of CTR, HPX, and HPX+EVs overlap. (**E**) The mitochondrial mean fluorescence intensities (MFI) of MitoTracker™ Orange were quantified from the groups indicated on the abscissa. Scatter plots represent determinations in which median values of fluorescence were recorded. In (**B**,**E**), differences were assessed using one-way ANOVA followed by Tukey’s test. (**B**,**E**) *** *p* < 0.001; **** *p* < 0.0001; NS: not significant.

**Figure 3 ijms-23-02906-f003:**
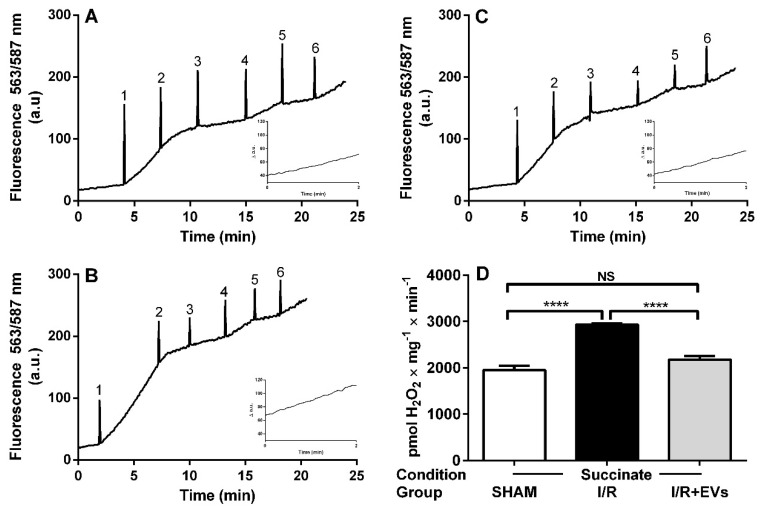
Generation of O_2_^•−^ in renal mitochondria 24 h after ischemia/reperfusion (I/R). Representative time courses of H_2_O_2_ formation from the O_2_^•−^ generated in renal cortex corticis mitochondria during different respiratory states, which were elicited by successive additions of substrates/inhibitors. (**A**) SHAM rats. (**B**) I/R conditions. (**C**) I/R + EVs; EVs were injected at the beginning of reperfusion. The numbers (spikes) indicate additions to the medium. 1:10 mM succinate; 2:0.1 mM ADP; 3:1 mM ADP; 4:0.2 μg/mL oligomycin; 5:1.5 μM FCCP; 6:2.5 μM Antimycin A. The insets allow a better comparison of the rate of formation of H_2_O_2_ over 2 min after the addition of succinate. The dismutation reaction 2O_2_^•−^ + 2H^+^ →H_2_O_2_ + O_2_ was non-limiting because an excess of superoxide dismutase (SOD) (60 U/mL) was added to the reaction medium. Standard curves were obtained by successive additions of 10 nM H_2_O_2_ pulses at 2 min time intervals, thereby allowing calculation of rates in pmol H_2_O_2_ × mg^−1^ × min^−1^ (see the following figure). (**D**) Quantification of the H_2_O_2_ formation after addition of succinate (insets). Bars are mean ± SEM; *n* = 7 (SHAM); *n* = 7 (I/R); *n* = 5 (I/R+EVs). Means were compared using one-way ANOVA followed by Tukey′s test. **** *p* < 0.0001; NS: not significant.

**Figure 4 ijms-23-02906-f004:**
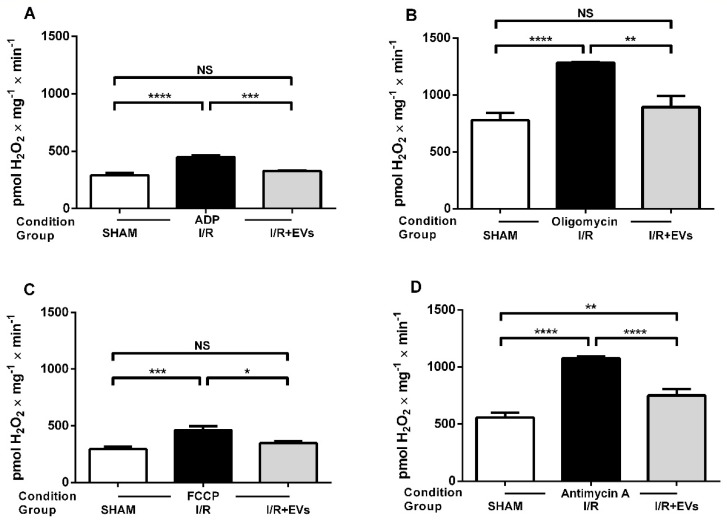
Quantification of H_2_O_2_ formation (energization with succinate) 24 h after I/R. Assays were carried out in the different respiratory states indicated on the abscissae. (**A**) Phosphorylating (1 mM ADP). (**B**) Non-phosphorylating (oligomycin). (**C**) Uncoupled (FCCP). (**D**) Residual (Antimycin A). Experimental groups are also indicated on the abscissae. Bars indicate mean ± SEM of the number of determinations indicated in the legend to Figure 3. * *p* < 0.05; ** *p* < 0.01; *** *p* < 0.001; **** *p* < 0.0001; NS: not significant (one-way ANOVA followed by Tukey´s test).

**Figure 5 ijms-23-02906-f005:**
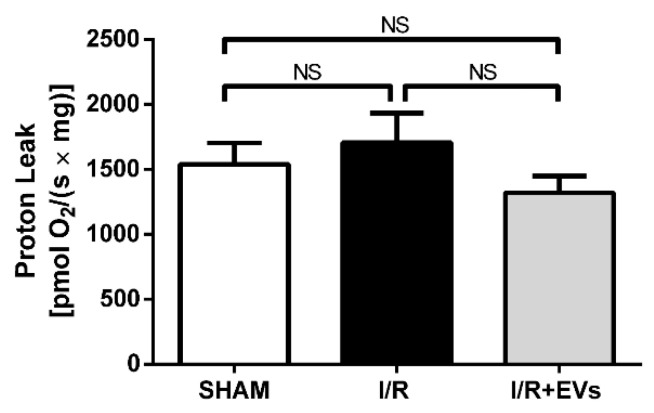
Proton leak (QO_2_ measured in the presence of oligomycin). The reaction medium was that used to determine the production of reactive O_2_ species in Figure 3, without addition of the components required for the formation of H_2_O_2_. The bars correspond to mean ± SEM of 6 (SHAM and I/R) and 5 (I/R + EVs) determinations with different mitochondrial preparations. Groups are those indicated on the abscissae. Not significant differences (NS) were found among the 3 groups (one-way ANOVA followed by Tukey´s test).

**Figure 6 ijms-23-02906-f006:**
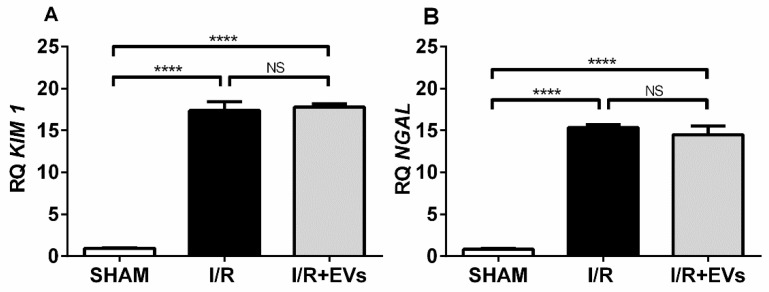
The expression (relative quantification) of the renal injury biomarkers 24 h after I/R are not modified by EVs. (**A**) *KIM-1*. (**B**) *NGAL*. The expression of the genes was investigated from total RNA extracted from cortex corticis. The bars represent mean ± SEM that were compared using one-way ANOVA followed by Tukey’s test. The experimental groups are indicated on the abscissae; *n* = 4 (SHAM), *n* = 7 (I/R), *n* = 5 (I/R + EVs), for both biomarkers. **** *p* < 0.0001; NS: not significant.

**Figure 7 ijms-23-02906-f007:**
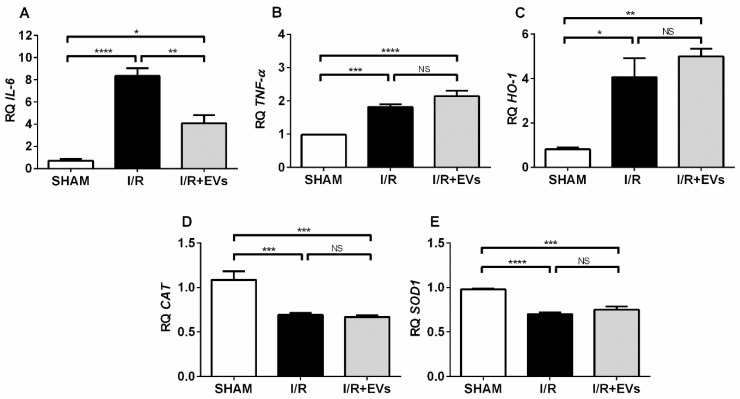
Expression of pro-inflammatory cytokines and antioxidant enzymes, of which expression was investigated from the total RNA extracted from cortex corticis. (**A**) *IL-6*. (**B**) *TNF-**α*. (**C**) *Heme oxygenase-1*. (**D**) *Catalase*. (**E**) *Superoxide dismutase-1*. Bars represent mean ± SEM that were compared using one-way ANOVA followed by Tukey’s test. Groups are indicated on the abscissae; *n* = 4 (SHAM), *n* = 7 (I/R), *n* = 5 (I/R+EVs). * *p* < 0.05; ** *p* < 0.01; *** *p* < 0.001; **** *p* < 0.0001; NS: not significant.

**Figure 8 ijms-23-02906-f008:**
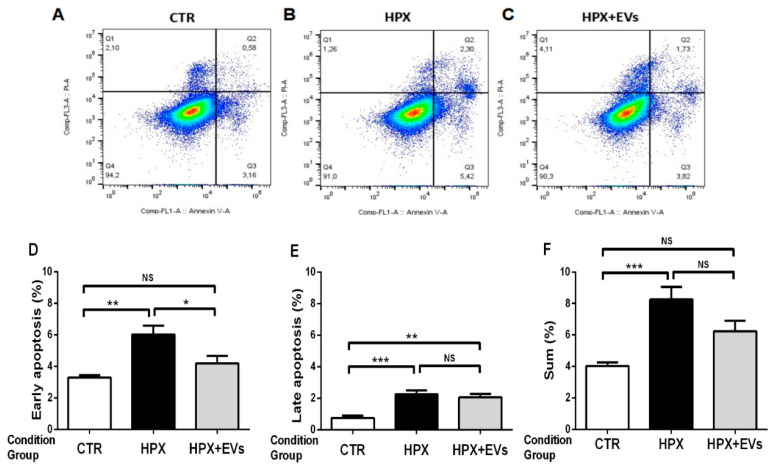
Co-culture with EVs decreases apoptosis of a lineage of proximal tubule cells (HK-2 line) subjected to hypoxia. Representative cytometry analysis of 3 × 10^5^ HK-2 cells cultivated in 1.5 mL of DMEM without serum for 24 h. (**A**) Cultures in normoxia (CTR). (**B**) Hypoxia (1% O_2_) (HPX) in the absence of EVs. (**C**) HPX in the presence of 2 × 10^9^ EVs (HPX+EVs). Then the cells were incubated for 24 h under 21% O_2_ in the same medium. (**D**–**F**) Quantification of cell death is expressed as a percent of total cells. (**D**) Early apoptosis, corresponding to ANX^+^PI^−^ cells. (**E**) Late apoptosis, corresponding to ANX^+^PI^+^ cells. (**F**) Sum of the cells that suffered early and late death: ANX^+^PI^−^ cells + ANX^+^PI^+^ cells. Bars represent mean ± SEM of different cultures in the conditions CTR (*n* = 16), HPX (*n* = 14), and HPX+EVs (*n* = 9). * *p* < 0.05; ** *p* < 0.01; *** *p* < 0.001; NS: not significant (one-way ANOVA followed by Tukey´s test).

**Figure 9 ijms-23-02906-f009:**
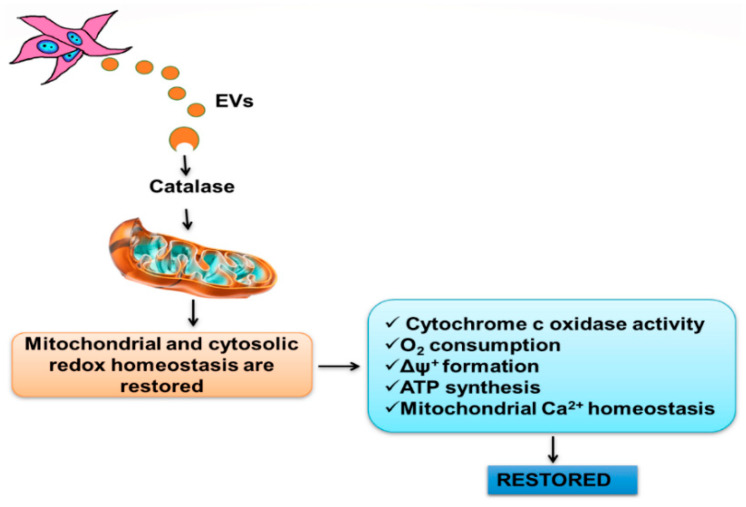
Suppressing the excess of O_2_^•−^: Proposed mechanisms for the rapid effects of EVs (24 h of reperfusion after renal ischemia). Mesenchymal cells (MSCs) secrete EVs that, after subcapsular administration and diffusion into the renal parenchyma [10], reach the tubular segments injured by I/R. By releasing several factors, including the catalase they carry [58], they contribute to maintaining the normal local redox state existing in the absence of injury. With mitochondrial and cytoplasmic redox homeostasis restored, the mitochondrial processes required for ATP synthesis are preserved and, therefore, the appropriate ATP supply is preserved for the transport demands and maintenance of tubular structures.

**Table 1 ijms-23-02906-t001:** List of the primers used in qRT-PCR analysis.

mRNA	Sequence
rSOD1 F1	AAGAGAGGCATGTTGGAGACC
rSOD1 R1	ACGGCCAATGATGGAATGCT
rCAT F1	CAGCTCCGCAATCCTACACC
rCAT R1	GGACATCGGGTTTCTGAGGG
rIL6 F1	AAGCCAGAGTCATTCAGAGC
rIL6 R1	GTCCTTAGCCACTCCTTCTG
rTNFA F1	CTTCTCATTCCTGCTCGTGG
rTNFA R1	TGATCTGAGTGTGAGGGTCTG
rKIM1 F1	ACCTGATCAGACAGAGTGTGC
rKIM1 R1	ATCTACAGAGCCTGGAAGAAGCA
rHO-1 F1	AGGTGCACATCCGTGCAGAG
rHO-1 R1	CTTCCAGGGCCGTATAGATATGGTA
rNGAL F1	GGGCTGTCCGATGAACTGAA
rNGAL R1	CATTGGTCGGTGGGAACAGA
rGAPDH F1	GCCAAAAGGGTCATCATCTC
rGAPDH R1	GGCCATCCACAGTCTTCT

## Data Availability

The raw data supporting the conclusions of this article will be made available by the authors, without undue reservation.

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
