# Peer review of "Early Effects of Extracellular Vesicles Secreted by Adipose Tissue Mesenchymal Cells in Renal Ischemia Followed by Reperfusion: Mechanisms Rely on a Decrease in Mitochondrial Anion Superoxide Production"

_ijms, 2022, doi:10.3390/ijms23062906_

Round 1

Reviewer 1 Report

1) In the abstract, the authors specified: “We conclude that the mechanisms by which EVs recover the renal structure and function after I/R are related to the normalization of the mitochondrial redox environment”. However, it is not clear how exactly extracellular vesicles improved mitochondrial redox environment and what renal structures were recovered.

2) The aims and objectives of the present study should be presented as an individual paragraph in the introduction section. The authors reported aims of the study in one paragraph, the objectives in another paragraph, all combined with the results and design of their previous study. Therefore, is hard to follow what aimed the authors to do in the present study.

3) Figure 8 contains only five parts: A-E, but in the legend of the figure it was also reported a F part: “Sum of the cells that suffered early and late death”

4) Although early apoptosis was reduced in the group with hypoxia and extracellular vesicles when compared to hypoxia alone, it was not reported any significant differences in late apoptosis between the two groups. Thus, the overall benefit of extracellular vesicles could be limited. It would be interesting an authors’ opinion regarding this issue.  

5) Conclusions should be reported as an individual section, not as a part of discussions.

6) Some phrases should be corrected:

-“The animals received EVs (I/R+EVs group) or saline, I/R group)” – (I/R group)

-“Acute kidney injury (AKI) is one of the more severe systemic syndromes in internal 57 medicine” – one of the most

Author Response

Please, see the atached file containing our response to Reviewer 1.

Reviewer 2 Report

Dear Editor,

The manuscript [IJMS] Manuscript ID_ ijms-1602176 entitled: “Early Effects of Extracellular Vesicles Secreted by Adipose Tissue Mesenchymal Cells in Renal Ischemia Followed by Reper-
fusion: Mechanisms Rely on the Restoration of the Redox Tissular Environment” by authors: Jarlene A. Lopes, Federica Collino, Clara Rodrigues-Ferreira, Luzia da Silva Sampaio, Glória Costa-Sarmento, Camila H.C. Wendt, Fernando P. Almeida, Kildare R. Miranda, Tais H. Kasai-Brunswick, Rafael S. Lindoso and Adalberto Vieyra offers good designed experiments considering characterization the early effects of extracellular vesicles secreted by adipose tissue mesenchymal cells in renal ischemia followed by reperfusion, as well as mechanisms rely on the restoration of the redox tissular environment.

In this work authors investigated the effects of early effects of EV on AKI and concluded that

General comments:

Title

The title is appropriate, precise and clear for readers.

Abstract

Written clearly and understandably. Includes all the elements for understanding what is written in the manuscript.

Introduction

The introduction is written concisely and clearly with the most basic facts concerning AKI. The hypothesis and goal of the paper are clearly defined.

In my opinion, the redox mechanisms and the influence of the super anion radikals on the appearance of AKI should be explained in more detail. Generally, more details about oxidative stress and AKI.

Materials  and Methods

All animal experiments and treatments have been approved by the appropriate institutions. All operating procedures are carried out with appropriate and standard techniques. All additional and auxiliary parameters related to the main goal of the work are carefully and correctly selected. All procedures used, such as isolation and characterization of extracellular vesicles secreted by adipose mesenchymal stromal cells, renal cells cultures and hypoxia/reoxygenation, flow cytometry, analysis of mitochondrial morphology and,evaluation of the electrical potential difference across the inner mitochondrial membrane from renal cells in co-culture with extracellular vesicles, time course of reactive O2 species formation, oxygen consumption by isolated mitochondria from kidney proximal tubule cells, RNA isolation, reversal transcription and real time quantitative polymerase chain reaction (qRT-PCR) are conducted using contemporary and adequate methods. Appropriate and adequate statistical analysis for data analysing were used.

Results

Obtained results show that that the EVs protect the renal structure and function after I/R and normalized of the mitochondrial redox environment. Authors shown that antioxidant components of EVs are central in the preservation in early processes of mitochondrial damage and cell death after I/R.

At the same time graphical and tabelar quality of presentation of the results in the text is at a high level.

Discussion and Conclusions

Discussion is in accordance with the obtained results and explains the obtained results in a logical way. Consequently, the conclusions are in line with the obtained results and the presented discussion.

There are no specific comments in the text.

Suggestion

The only suggestion I would make is that lately some of newest references which explain the key molecular mechanisms of protective action of some substances on AKI did not mentioned in the text and in my opinion would be very useful in interpreting and quality improvement of the results, for example:

  1. Đurašević S., Stojković M., Bogdanović Lj., Pavlović S., Borković-Mitić S., Grigorov I., Bogojević D., Jasnić N., Tosti T., Đurović S., Đorđević J., Todorović Z. (2019). The effects of meldonium on the renal acute ischemia/reperfusion injury in rats. J. Mol. Sci., 20, 5747; DOI:10.3390/ijms20225747

Conclusion of the Reviewer

In my opinion, this is a very high quality manuscript and I suggest to the editor to accept this MS [IJMS] Manuscript ID_ ijms-1602176 for publication in this form with the proproposed suggestions.

General conclusion: Acceptable for publication.

Author Response

Please, see the attached file containing our response to Reviewer 2.

Reviewer 3 Report

In this article the author investigated many things such as

Intense anion superoxide production in acute renal injury
Mitochondria are the main source of anion superoxide
Extracellular vesicles secreted by mesenchymal cells restore normal redox
Intravesicular catalase and Heme oxygenase-1/Nuclear factor erythroid

But no one if fully supported by full set of experiments.

Fig-1 describes EV characterization, but how you validate that these are adipose tissue derived EVs and not any other source.

Please provide the raw image of EVs for electron microscopy in supplemental files.

Figure 2A, Is the scale bar and mitochondrial size correct??? Again please add raw image in the supplemental files.

Author Response

Please, see attached file containing our response to Reviewer 3.

Round 2

Reviewer 1 Report

no further comments

Reviewer 3 Report

No comments